# Design of a Functional Eye Dressing for Treatment of the Vitreous Floater

**DOI:** 10.3390/jpm12101659

**Published:** 2022-10-05

**Authors:** Wen-Shuang Fan, Shuan-Yu Huang, Hong-Thai Nguyen, Wen-Tsung Ho, Wen-Hung Chao, Fen-Chi Lin, Hsiang-Chen Wang

**Affiliations:** 1Department of Ophthalmology, Dalin Tzu Chi General Hospital, 2, Min-Sheng Rd., Dalin Town, Chia-Yi 62247, Taiwan; 2Department of Optometry, Central Taiwan University of Science and Technology, No.666, Buzih Road, Beitun District, Taichung City 406053, Taiwan; 3Department of Mechanical Engineering, Advanced Institute of Manufacturing with High Tech Innovations (AIM-HI) and Center for Innovative Research on Aging Society (CIRAS), National Chung Cheng University, 168, University Rd., Min Hsiung, Chia Yi 62102, Taiwan; 4Master’s Program in Wellbeing Technology and Biomedical Engineering, Yuanpei University of Medical Technology, No.306, Yuanpei Street, Hsinchu 30015, Taiwan; 5TO2M Corporation, 5F., No. 2, Kedong 3rd Rd., Zhunan Township, Miaoli County 35053, Taiwan; 6Department of Ophthalmology, Kaohsiung Armed Forces General Hospital, 2, Zhongzheng 1st.Rd., Lingya District, Kaohsiung City 80284, Taiwan

**Keywords:** vitreous floaters, functional eye dressing, eye floaters, oxygen therapy, hydrogen therapy

## Abstract

With the rapid development of display technology, related diseases of the human eye are also increasing day by day. Eye floaters are one of the diseases that affect humans. Herein, we present a functional ophthalmic dressing that can permeate the skin tissues of the eyes through oxygen and hydrogen to improve the symptoms of floaters. In clinical tests, the symptoms of sensory floaters improved in 28 patients, and the recovery rates of mild, moderate, and severe floaters were about 70%, 66.7%, and 83.3%, respectively.

## 1. Introduction

In the past, 90% of vitreous floaters were caused by aging [1]. With the development of 3C (computer, consumer electronics, and communication) products and the increase in myopia, 15% of people aged 20–29 years will develop lymphadenopathy, indicating the gradual acceleration of the eye-aging rate. Lymphadenopathy refers to the presence of floating objects in the eye caused by thickened clumps of vitreous jelly [2,3,4,5,6,7]. After the age of 40, the vitreous will become dense and gel-like, in which some denser bundles of collagen fibers form. For people with high myopia, the anterior–posterior diameter of the eye axis is stretched, destroying the dense structure of the vitreous and leading to vitreous degeneration and turbidity. When light is projected to the eyeball, it passes through the interface of the thicker fiber bundle. The vitreous jelly blocks light coming from the front of the eye and casts shadows on the retina. This phenomenon is called degenerative floaters [8,9,10,11,12,13,14].

Pathological floaters are another type of vitreous floater. Pathological floaters occur when there is a sudden increase in floaters or when the light and shadow of lightning can be observed. Although pathological floaters only account for 5% of floaters, they may seriously harm eyesight. The causes include retinal tears, accompanying retinal detachment, bleeding caused by retinal vein vascular occlusion, diabetic retinopathy, and macular degeneration, which causes blood to flow into the vitreous. At present, the diagnosis of pathological floaters depends on the experience of ophthalmologists. Primary vitreous floaters are mainly produced by the intrinsic structure of the vitreous, and the stacked collagen fiber bundles form visible fibers, which first appear in the central vitreous, where they have a linear structure. With age, they become more and more thick and irregular and are common in young axial myopia. In old age, the glass body liquefies and forms a cavity, and light scattering from the glass cavity wall may cause a floating phenomenon [15,16,17,18,19]. Secondary floaters are exogenous turbidity in the vitreous that is usually composed of protein, amyloids, or cells. However, the most common cause of secondary vitreous floaters is anterior or vitreous hemorrhage, which can cause a sudden onset of floaters and blurred vision [20,21]. Inflammatory diseases (infectious or non-infectious) or malignant tumors (such as lymphoma) can cause an increase in vitreous cells. If the number is large, it may also cause symptomatic vitreous floaters.

The currently available treatments for floaters include vitrectomy and Nd: YAG laser vitrectomy. Many studies have evaluated the success and potential risks of vitrectomy for vitreous floaters [22,23,24,25,26,27,28]. Although vitrectomy is an effective treatment method, it may affect the lens and accelerate the occurrence or deterioration of postoperative cataracts. On the other hand, Nd: YAG laser vitreous dissolution is used to destroy the vitreous collagen fiber bundles. It only treats the fiber bundles that are relatively far from the retina. However, it only destroys the aggregation and cannot remove the aging cells from the body. It is better for larger aggregations. Nd: YAG laser vitreous dissolution using newer technology has been proven to be safe and effective in treating symptomatic Weiss ring vitreous floaters. The primary outcome measures were the percentage of symptom improvement after treatment and the rate of postoperative complications. In the study, Delaney et al. [23] reported that the Nd: YAG vitreous improved symptoms in only one-third of patients. Furthermore, clinical improvement was only moderate, subjectively graded to no greater than 50% among 93.3% of patients. Only 38% of the 39 eyes with vitreous floaters showed symptomatic improvement moderately after Nd: YAG laser treatment; not only that, the symptoms could not be improved in terms of complete vision resolution. Moreover, this method of laser therapy also causes complications that are not completely reported. Specifically, complications after surgery can be mentioned as complications of retinal detachment, focal lens opacities, and minor retinal hemorrhages. The exact pathogenesis of this complication is unknown but may involve shock wave generation, vitreous disturbances, or the formation of cellular debris or inclusions. A study by Nguyen et al. [25] proposed a method to evaluate the recovery of patients with vitreous floaters after treatment with Nd: YAG. Evaluation methods are based on visual function questionnaires, comparative statistical methods are based on quantitative ultrasonography index and contrast sensitivity. There were 38 eyes with vitreous floaters status examined after performing Nd: YAG treatment. Of these, 13 cases showed improvement in symptoms after surgery. However, there were still 25 cases who feel unsatisfied with the results of vision therapy. Another study by Souza et al. [26] reported no side effects or a recurrence of vitreous floaters. The study used color photo imaging to evaluate YAG laser vitreolysis for symptomatic vitreous floaters. A total of 32 patients were participating in the survey based on the visual function questionnaire. After 6 months of follow-ups, color imaging showed improvement in vitreous opacity over time in 93.7% of study eyes. In trials, symptomatic amorphous posterior clinical vitreous floaters were detected by a novel optical coherence tomography (OCT) [27] and successfully treated with a YAG laser optimized for vitreous dissolution. A study by Landa et al. [27] has shown that the spectral domain OCT can assist in the diagnosis of retinal detachment that cannot be excluded only on clinical examination. In general, laser therapy, particularly Nd: YAG, was found to be more effective and safer in the treatment of vitreous floaters. OCT is one method to evaluate the results after treatment is relatively effective. However, the limitation of laser therapy is that it brings many complications and the success rate of vision recovery is not high. The reason may be that this is an invasive method to the vitreous in the retina, causing disturbances in the retinal environment after treatment, and leading to complications. This means that laser treatment is considered when used as a method of vitrectomy.

In this study, we propose a functional dressing for eye floaters as a non-surgical, semi-invasive treatment with no direct impact on vitreous humor. We demonstrate a functional dressing, which has a structure that effectively allows oxygen and hydrogen to penetrate the skin. After the dressing is attached to the skin, it can generate hydrogen and oxygen through moisture. The function of oxygen is to increase blood oxygen, and the function of hydrogen is to fight inflammation and cell apoptosis in the eye and inhibit the production of excessive active oxygen. The dressing uses hydrogen and oxygen to penetrate tissues, and protect wounds from infections, creating a favorable environment for the recovery of lesions around the eye area. We propose a technique to evaluate post-treatment outcomes by analyzing OCT images, providing a more advanced visual assessment than traditional assessment methods through the visual functioning questionnaire system or other measures such as quantitative ultrasonography or contrast sensitivity.

## 2. Materials and Methods

### 2.1. Oxygen Therapy

The purpose of oxygen therapy is to treat hypoxia and reduce the clinical symptoms caused by hypoxia. Reactive oxygen species (ROS) are small molecules derived from oxygen and are a by-product of biological aerobic metabolism. They can be used as oxidants or are easily converted into oxygen-free radicals, which are one of the most important elements in oxygen therapy. They react with a variety of molecules, including other small organic molecules such as carbohydrates, lipids, proteins, and nucleic acids. However, excessively high levels of ROS can cause damage to the cell and gene structure. Usually, cells pass enzymes (e.g., superoxide dismutase) to reduce the damaging effect of ROS on cells. Oxygen therapy is also used to treat some patients with chronic oxygen deficiency, such as patients with chronic obstructive pulmonary disease or cystic fibrosis. On the other hand, oxygen is needed in every synthetic action of wound repair, including the synthesis of adenosine triphosphate, collagen, protein, and phagocytes, as shown in Figure 1a. Nitrogen oxides (NOx) enzyme is the main source of ROS. The production of ROS during wound healing is essential for cell-signal transduction, angiogenesis, and wound disinfection. Oxygen generates superoxide anions through the catalysis of NOx to generate hydrogen peroxide and, subsequently, undergoes the action of redox signals until angiogenesis, as shown in Figure 1b. In addition, as shown in Figure 1c, hyperbaric oxygen therapy can increase the level of vascular endothelial growth factor in the wound and monoxide and enzymes in the bone, allowing vascular endothelial stem cells to return to the wound. The ischemic site merges into new blood vessels [29]. After receiving oxygen, nitric oxide (NO) and enzymes will be affected by enzyme catalysis to synthesize NO and help regulate blood vessel tension and angiogenesis, as shown in Figure 1d. In the human eye, supersaturated oxygen emulsion can also be used for the local treatment of ocular trauma. Oxygen therapy can improve limbal ischemia, accelerate the formation of corneal epithelium, increase corneal transparency, and reduce corneal blood vessel formation [30].

### 2.2. Hydrogen Therapy

Hydrogen is one of the most common substances in nature. It can selectively neutralize cytotoxic ROS and reduce inflammation. Hydrogen is used in a variety of medical applications. At present, the concentration of H_2_ in the air does not exceed 4%. Low-H_2_ concentrations exhibit therapeutic effects on local inflammation of the eyes, ears, nose, and liver, as well as pancreatitis, systemic inflammatory syndrome, sepsis, and neurodegenerative diseases. In 1975, high concentrations of H_2_ were found to inhibit tumor growth [31]. Hydrogen can play a protective role in various ROS-related diseases, including reducing bowel transplant damage in organ transplantation, treating chronic inflammation, and reducing ischemia-reperfusion injury. Hydrogen is also used to treat various ocular diseases, especially in the retina, which is a place where oxygen is highly needed. However, if excessive free radicals accumulate and increase oxidative pressure, they will peroxidize the lipids in the retina, causing retinal hypoxia. Once hypoxia will cause the production of new blood vessels, hydrogen molecules play a very important role at this time. H_2_ is a perfect anti-free radical, and it can protect the retina from vascular proliferation. Oxidative stress triggers the development of a variety of human diseases and injuries, including eye diseases. The human body will induce an oxidative stress response due to excessive production of ROS or reduced production of antioxidants. In order to replace these weakened antioxidants, substances with effective antioxidant properties are needed to inhibit oxidative stress and promote healing. Molecular hydrogen (H_2_) is very suitable for this purpose due to its unique properties. H_2_ is the only antioxidant that crosses the blood–brain barrier and the blood–eye barrier. Due to its small molecular weight, it can quickly penetrate tissues and effectively remove active oxygen. H_2_ mainly removes hydroxyl free radicals and peroxynitrite. In addition to its antioxidant effect, H_2_ also has anti-inflammatory, anti-apoptotic, cell protection, and mitosis effects. Even when used at high concentrations, H_2_ still maintains its non-toxic properties. Figure 2 shows the biological effect mechanism of hydrogen. The main molecular target of the biological effect of H_2_ is ROS. The effects on chronic inflammation, signal transduction, genes, immunity, and metabolism (mitochondria) are essentially exerted through ROS. Exogenous damage caused by radiation and other factors induces excessive cellular ROS production [31]. H_2_ penetrates the biomembrane and effectively reaches the cell nucleus. H_2_ will selectively remove OH and ONOO−, thereby, preventing DNA damage, as shown in Figure 2a. H_2_ reduces the number of apoptotic factors such as caspase-3, caspase-12, caspase-8, and Bax. Some regulatory apoptosis factors such as Bcl-2 and Bcl-xL exhibit an increasing trend, making human cells have anti-apoptotic effects, as shown in Figure 2b. In addition, H_2_ reduces the number of inflammatory cytokines such as interleukin (IL)-1β, IL-6, tumor necrosis factor-α, intercellular adhesion molecule-1, and high mobility group protein-1, leading to anti-inflammatory effects in human cells, as shown in Figure 2c. H_2_ also regulates the signal transduction within and between many pathways, but the exact target and molecular mechanism need to be further studied, as shown in Figure 2d. In general, H_2_ reduces the risk of oxidative stress related to lifestyle and environment by reacting with strong ROS in a cell-free reaction.

### 2.3. Functional Dressing

Oxygen is a vital source of energy necessary for cell repair and renewal. Once in the lungs, oxygen is carried to the cells through the blood in the capillaries. Due to hypoxia, epidermal cells appear inactive and hibernate, leading to the appearance of grayish-yellow and dark black oxidation marks and dryness, roughness, wrinkles, and sagging of the skin. Air pollution, dust, various toxic substances, and computer radiation are the most likely to cause cell hypoxia, which will cause skin cells to repair themselves, store water, and decrease their defenses. With the exception of supplemental oxygen, all other treatments are nearly futile as they only correct the symptoms but do not treat the root cause. Oxygen not only plays an important role in maintaining the health of our skin but also in wound healing and preventing infection. Oxygen can effectively increase the speed of wound healing. In a low-oxygen environment, healing might be delayed. By contrast, healing is accelerated and the risk of infection is greatly reduced in a high-oxygen environment. In addition to wound healing and infection prevention, oxygen can also be in the form of active oxygen, which can effectively reduce the production of melanin and scars and improve skin tone. Figure 3 shows the structure of the functional dressing used in the study. The functional dressing is from TO2M Co. (Miaoli County, Taiwan), model BXX01. This dressing contains sodium peroxide, sodium hydroxide, aluminum powder, and oxalic acid. It generates hydrogen by adding water [32,33]. It can not only protect the wound but also combine oxygen (O_2_) and hydrogen (H_2_) to provide an ideal environment for the affected area, block foreign bodies, and reduce the chance of infection.

### 2.4. Type B Ultrasonic Scanner (Nidek RS-3000)

Type B ultrasonic scanner includes choroidal mode, which provides a comprehensive evaluation for choroid, retina, and glaucoma analysis. The principle of OCT scanning is to use the spectral domain of OCT with a scan range *X* axis is 3–12 mm, the *Y* axis is 3–9 mm, and the *Z* axis is 2.1 mm, using an 880 nm Super-Luminescent Diode light source. Scan speed can achieve up to 85,000 A-scans/s with averaging 120 images. The device can perform retina analysis, glaucoma analysis, angio-scan, or real-time compensation for eye movements which ensures higher image quality and maximum reproducibility. The advanced mode of RS-3000 can be used for measurement with ultra-low sensitivity depending on the pathology to be evaluated. The 9 mm × 9 mm wide-area scan ensures excellent coverage of the entire retinal structure. The unique Eye Tracer technology can use fundus information obtained from high-definition images for precise measurement. The Eye Tracer technology combines positioning, tracking, and automatic shooting functions for convenient and rapid measurement. During the macular line scan, micro-tracking and other involuntary eye movements can be compensated by the “Tracking HD” function. This function ensures that up to 120 macular scan images are aligned to enhance image averaging. Subsequent images are precisely aligned with the baseline data to achieve high reproducibility. The automatic registration function can compensate for the knob in the image acquisition process, thereby, improving the quality of subsequent data. The B-mode ultrasound scanner is an important diagnostic and predictive tool to determine diseases at the back of the eyeball when obvious vitreous opacity is present. There have been many research trials using Scan-B Ultrasound to detect vitreous floaters. Oksala et al. [34] used ultrasound to detect echoes from vitreous collagens. Mamou et al. [35] effectuated similar experiments by analyzing the contrast sensitivity obtained from this ultrasound device. Hence, ultrasound is typically used to assess the diagnosis of vitreous floaters.

### 2.5. Symptoms of Vitreous Floaters

In the imaging of the B-mode ultrasound scanner, vertical and horizontal analysis diagrams inside the vitreous body and OCT assist the doctor to make more accurate judgments and treatments. Patients with mild clinical symptoms will see black lines floating and occasional flashes of tiny spots of light that may not affect vision. Under the analysis of the B-mode ultrasonic scanner, the turbid part of the vitreous is not obvious, but a slight shadow and blackness can still be seen on the vitreous. Patients with moderate clinical symptoms have increased dark shadows and light spots in front of their eyes (Figure 4a). In the imaging of the B-mode ultrasound scanner, local shadows begin to appear inside the vitreous, as shown in Figure 4b. When this symptom persists, the condition is worsening, and the patient must seek a medical facility for treatment. Finally, patients with severe and more significant turbidity have a large number of black shadows in front of their eyes, and their vision may even be reduced by varying degrees. Dust or thick flocculent blocks floating may be present near the vitreous body and the macula. As shown in Figure 4c, when the vision is severely affected, surgical treatment should be considered. Clinical symptoms and details of pathological signs diagnosed and consulted by doctors represented in OCT images are shown in Table 1.

## 3. Results and Discussions

Figure 5a–c show the measurement results of the ultrasound scan of patient A before and after treatment. In Figure 5a, before treatment, patient A has a darker shade in the center of the macula in the middle of the vitreous, and many tiny floating objects can be observed in the vitreous. In Figure 5b, the shadow of the fovea gradually fades after 2 months of treatment, and the floating objects are also reduced and lighter. In Figure 5c, 4 months after treatment, the shadow of the fovea is lighter, and the floating objects are almost disappeared. Figure 5d–f show the results of the ultrasound scanner measurement of patient B before and after treatment. In Figure 5d, the shadow covers almost the entire macula and is even more severe than the symptoms of patient A. The vitreous body is also full of many tiny floating objects. After 2 months of treatment, the shadow of the macula is significantly lighter, the treatment effect has not decreased due to more serious symptoms, and the number of floating objects is reduced (see Figure 5e). After 4 months of treatment, the macular shadows and floating objects almost completely disappear, as shown in Figure 5f. The diagnosis of vitreous floaters through ultrasound scanners provides a fairly reliable basis. Type-B ultrasound systems are capable of assessing changes from the wave at interfaces of tissues with different densities, i.e., interfaces between liquid and gel vitreous or echoes from vitreous collagens. This was confirmed in some references that found that vitreous opacity in the vitreous vestibule was most correlated with diminishing contrast. When the amount of echoes from gel-liquid interfaces tend to decrease, it shows that the environment in the retina is gradually returning to equilibrium [24,34,35].

Recently, due to aging or excessive use of 3C products, glass floaters have continued to increase. When the vitreous body of the eye is chaotic, vitreous floaters cast shadows on the retina, making it seem like something is floating in front of the eyes [36]. In a previous clinical trial, 10, 12, and 6 subjects had mild, moderate, and severe floaters, respectively. After 2 months of treatment for patients with mild floaters, seven patients almost no longer had symptoms of floaters. However, some patients had bad eye habits. Three patients with mild symptoms had improper eye habits, such as using mobile phones in dark places, staying up late, and overuse of eyes, resulting in symptoms of floaters. After 2 months of treatment for patients with moderate floaters, eight patients had mild symptoms, and the other 4 are due to complications such as recurrent vitreous hemorrhage and corneal edema. As well as poor healing of the corneal epithelium, postoperative intraocular pressure increase, retinal tears, etc., continuous regular follow-up and treatment are required to achieve the therapeutic effect. However, new complications may continue to occur in the acute phase. Thus, 1 week after diagnosis, close follow-up visits are required within 3 months. If patients remain stable, regular visits will be made for 3–6 months. However, patients with severe floaters usually have pathological floaters, which are more likely to occur over 50 years of age. People with high myopia, diabetes, high blood pressure, cataract surgery, eye injury, and eye inflammation. Due to the rupture of blood vessels in the retina, retinal tears or detachment of the retina result in inflammation of the tissues around the vitreous body, causing a large amount of white blood cell suspension to leak out from the vitreous body. Usually, the flying mosquitoes seen by the patient are a thick black shadow, which seriously affects the vision and needs immediate treatment. After 2 months of clinical trial treatment, six patients had moderate symptoms of floaters. Among the six patients with severe floaters, the use of functional dressings can improve the treatment effect due to the relatively large volume of floating objects. Among these patients, five had obvious treatment effects. The floating objects were not only greatly reduced, but the poor vision was also slightly improved. However, floating objects may not be removed by a single treatment. Multiple treatments and continuous follow-ups are required, and the other one is due to the usual and continuous improper use of the eyes, which results in the lack of effectiveness of the treatment. Taken together, when patients see floaters in front of their eyes, they should visit the ophthalmology department for a thorough retinal examination of the fundus. Patients with mild symptoms of floaters may have these symptoms throughout their lives, but they still need to be monitored regularly by an ophthalmologist. After a period of adaptation, floaters usually leave the field of vision and gradually disappear. Patients can also try to move their eyes (looking up and down) to make the fluid in the eyes form fluctuations, which can temporarily make floaters disappear. However, pathological floaters caused by vitreous hemorrhage, peripheral retinal tears, and retinal detachment require more aggressive treatment to avoid further deterioration. The number of patients who recovered positively after 2 months of therapy is shown in Table 2.

## 4. Conclusions

The treatment of vitreous floaters is currently attracting doctors’ and researchers’ attention. There are many advanced solutions to this problem. Our method has shown the patient’s recovery effect in more than 4 months of intensive treatment. Compared with laser therapy, using a functional eye dressing has more potential treatment results and does not cause side effects with the ingredients in the device. Functional eye dressing offers a novel solution for semi-invasive therapy, providing potential therapeutic results and no side effects with ingredients in the device. Our method is portable, compact, and suitable for treatment in the eye and face area. The treatment results achieved for the three grades of vitreous floaters disease were 70%, 66.7%, and 83.3%, respectively. Moreover, the evaluation of treatment results visualized by analyzing through OCT images will bring essential assistance to doctors in the diagnosis and conclusion of treatment regimens. We hope that the use of functional eye dressing will be a new solution in the treatment of vitreous floaters.

## Figures and Tables

**Figure 1 jpm-12-01659-f001:**
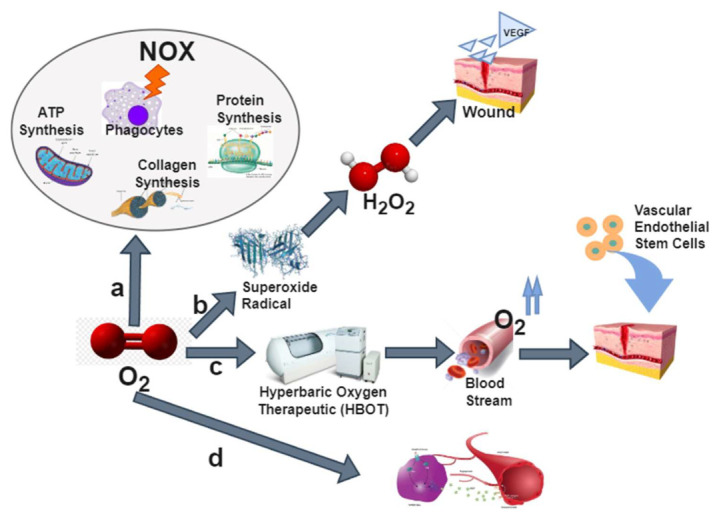
Oxygen repair cell derivation process. (**a**) Synthesis of adenosine triphosphate (ATP), collagen, protein, phagocytes, and oxygen. (**b**) Enzyme-catalyzed redox repair in the cell. (**c**) Hyperbaric oxygen therapeutic cell repair. (**d**) Synthesis of nitric oxide (NO) angiogenesis. Vascular endothelial growth factor (VEGF).

**Figure 2 jpm-12-01659-f002:**
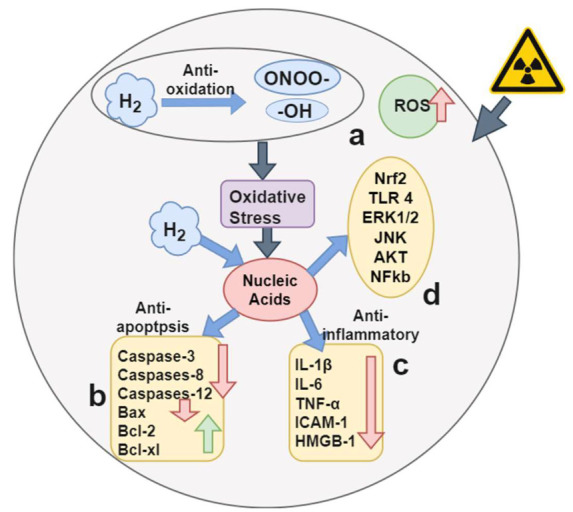
Biological effect mechanism of hydrogen. (**a**) H_2_ selective removal of -OH and ONOO−. (**b**) H_2_ stimulating human apoptosis factor cell regulation. (**c**) H_2_ promoting human inflammatory factor cell regulation. (**d**) Unknown molecular mechanism of H_2_ regulation.

**Figure 3 jpm-12-01659-f003:**
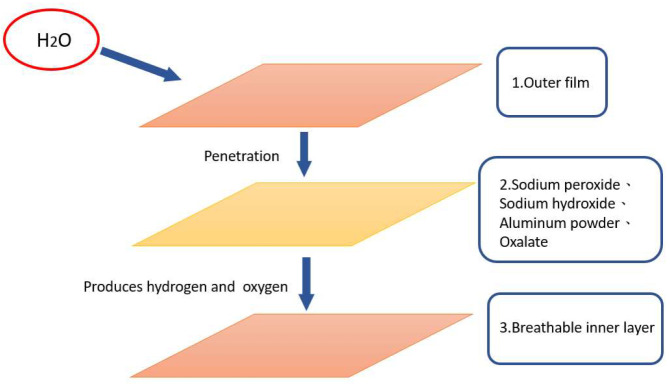
Structure of the functional dressing. The outer film layer has a structure that allows the penetration of H_2_O easily, the middle layer consists of chemical compounds that generate hydrogen and oxygen, and sequentially escape through the inner film layer.

**Figure 4 jpm-12-01659-f004:**
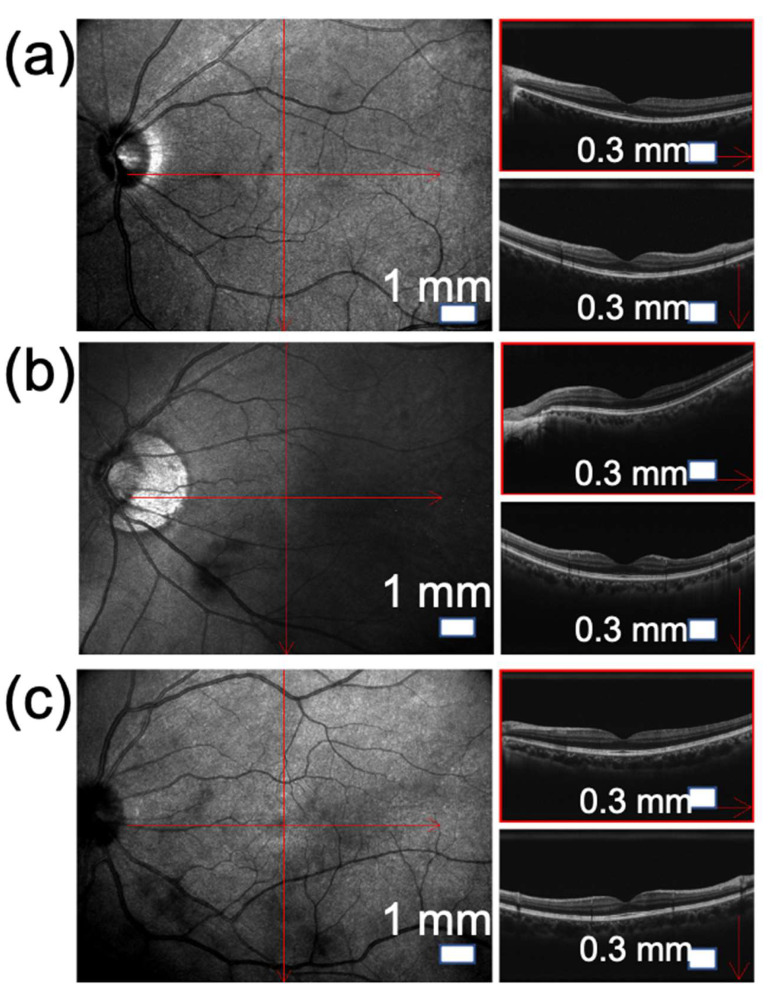
Measurement results of B-type ultrasound scanner. (**a**–**c**) Imaging analysis of mild, moderate, and severe floaters, respectively. The figure on the right is the OCT in the horizontal and vertical directions.

**Figure 5 jpm-12-01659-f005:**
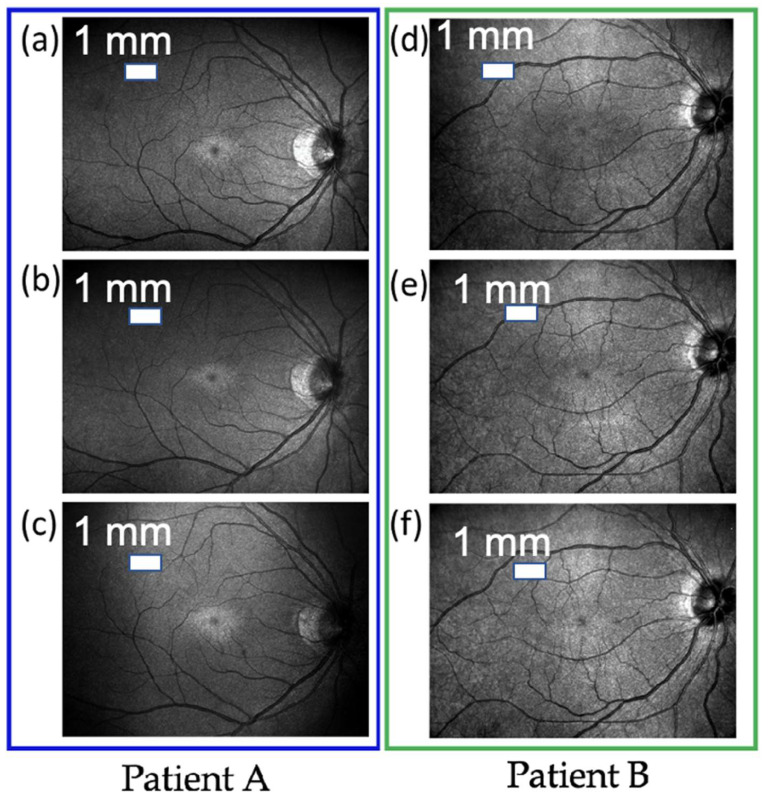
Comparative analysis of clinical treatment effects. Results showed improvement of eye floaters symptoms through fading of macular shadows and floating objects compared to baseline disease (**a**,**d**), after 2 months of treatment (**b**,**e**), and after 4 months of treatment (**c**,**f**) of 2 patients A and B.

**Table 1 jpm-12-01659-t001:** The clinical symptoms and pathological sign details are shown on OCT images for levels of vitreous floater.

Levels of Floaters	Mild	Moderate	Severe
Number of patients	10	12	6
Clinical symptoms	− Occasional black lines floating and flashes of tiny spots of light − Vision: short-term fatigue	− Black lines floating and flashes of tiny spots of light − Vision: short-term fatigue, increased myopia	− Appearance of lesions on the cornea, infection, itching − Vision: prolonged eye fatigue, increased myopia
Pathological signs shown in OCT	Slight shadow and blackness appear on the vitreous	Local shadows appear inside the vitreous	The shadows and blackness appear wider and darker

**Table 2 jpm-12-01659-t002:** The number of patients participating in clinical trials and the number of patients recovering actively after 2 months of treatment.

Levels of Floaters	Mild	Moderate	Severe
Number of patients	10	12	6
Number of recovery patients	7	8	6
Avarage recovery rate	70.0%	66.7%	83.3%

## Data Availability

The data presented in this study are available in this article.

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
