# Peer review of "Design of a Functional Eye Dressing for Treatment of the Vitreous Floater"

_jpm, 2022, doi:10.3390/jpm12101659_

Round 1
Reviewer 1 Report
Reviewer comments
Title: - Design of a functional eye dressing for treatment of the vitreous floater
This manuscript presents a functional ophthalmic dressing designed to improve the symptoms of floaters by permeating the skin tissues of the eyes with oxygen and hydrogen. To achieve the therapeutic effect, they use peroxides, hydroxides, solid acids, and aluminum powder to make the dressing absorb moisture in the air and then generate hydrogen and oxygen. To make the manuscript up to the level of the journal, I believe it needs a major revision.
1. A similar statement appears in the abstract of lines no. 24 and 28.
2. Ensure that the 28-patient data are presented correctly in a tabular format.
3. It is essential to summarise the results and present them properly.
4. Analyse the present study in relation to recent studies that have been conducted on vitreous floaters
5. The manuscript should also include this comparative study in tabular format.
6. There is room for improvement in the abstract and conclusion.
7. Overall, all technical details should be explained in a brief manner by the author.
8. The English language needs to be improved.
Reviewer 2 Report
After reading this manuscript, I feel that this manuscript is more like an experiment report than a good academic article. Only Figure 5 is the experimental result. The experiment results are not presented completely and the data is not analyzed well (no curve or data graphs, no tables, only image results). Based on the results shown in Figure 5 alone, the conclusions of the article are hardly convincing. Detailed comments are as follows:
(1) In the abstract, one sentence is repeated.
"We used peroxides, hydroxides, solid acids, and aluminum powder to make the dressing absorb moisture in the air and then generate hydrogen and oxygen to achieve the therapeutic effect."
(2) What is the novel optical retinal tomography in line 74? Note that "OCT" is an abbreviation for "optical coherence tomography". If optical retinal tomography is a novel technology not exactly the same as optical coherence tomography technology, please add the relevant references.
(3) In the introduction part, the authors introduce laser therapy for floater diseases. However, it does not have a literature review on something related to functional eye dressing therapy for floater diseases. What is the novelty of this study distinguished from others in this field? The authors should talk about what others have done and have not done in the field of drug vitrification therapy especially using functional eye dressing to treat floater diseases.
(4) Figure 1 is not clear. The words on the picture are elongated and some are indistinguishable.
(5) Figure 4 and 5 are without scale bars.
Round 2
Reviewer 1 Report
No comments.
Author Response
We appreciate your review.
Reviewer 2 Report
Thanks to the author for the effort and following my suggestions. The manuscript is much better now. Most of my concerns are solved, except for adding the discussion on what others have done and have not done in the field of drug vitrification therapy especially using functional eye dressing to treat floater diseases. This is very essential for readers to understand what this work distinguishes from others and why this work is important. The authors deleted the introduction of laser therapy for floater disease. The authors void the important and dwell on the trivial. This does not solve the problem.
